# High Prevalence and Clinical Associations of Vitamin D Deficiency in Inflammatory Bowel Disease: Evidence from a Tertiary Center Cohort

**DOI:** 10.3390/nu17233698

**Published:** 2025-11-25

**Authors:** Theodora Kafentzi, Ploutarchos Pastras, Ioanna Aggeletopoulou, Efthymios P. Tsounis, Georgios Geramoutsos, Nikitas Kimiskidis, Maria Bali, Konstantinos Thomopoulos, Georgia Diamantopoulou, Georgios Theocharis, Christos Triantos

**Affiliations:** Division of Gastroenterology, Department of Internal Medicine, University of Patras, 26504 Patras, Greece; dora_kafentzi@hotmail.com (T.K.); ploutarchosp96@gmail.com (P.P.); iaggel@upatras.gr (I.A.); makotsouno@gmail.com (E.P.T.); giorgosgeramoutsos@gmail.com (G.G.); nkimiskidis.93@gmail.com (N.K.); mariabalimed@gmail.com (M.B.); kxthomo@hotmail.com (K.T.); geodiamant@hotmail.com (G.D.); georgiostheocharis@hotmail.com (G.T.)

**Keywords:** vitamin D deficiency, inflammatory bowel disease, biologic therapy, systemic inflammation, platelets, SGOT

## Abstract

**Background/Objectives:** Vitamin D in its active form, 1,25-dihydroxyvitamin D_3_ [1,25(OH)_2_D_3_], plays a critical role in immune regulation, gut barrier function, and systemic inflammation. Its deficiency is frequent in Inflammatory Bowel Disease (IBD), but the clinical implications remain uncertain. The aim of the study is to assess the prevalence of vitamin D deficiency in a well-characterized IBD cohort in Western Greece, and explore its associations with clinical features, laboratory biomarkers, and treatment intensity. **Methods:** In this cross-sectional study, 184 consecutive, well-characterized IBD outpatients followed at a tertiary referral center in Western Greece underwent clinical evaluation and laboratory testing between January 2023 and December 2024. Vitamin D is determined by measuring 25-hydroxyvitamin D [25(OH)D], which reflects the body’s vitamin D stores due to its longer half-life compared with the biologically active form. Deficiency was defined as serum 25(OH)D < 20 ng/mL. Associations with disease type, clinical and laboratory biomarkers, severity indices, and treatment were analyzed using multivariate logistic regression. **Results:** Vitamin D deficiency was identified in 67 patients (36.4%). Although unrelated to disease type, hospitalization, surgery, or disease activity indices, deficiency correlated with systemic inflammation, nutrition/metabolic markers, and treatment intensity. More specifically, vitamin D-deficient patients exhibited higher platelet counts (*p* = 0.005) and erythrocyte sedimentation rate (ESR) (*p* = 0.014), lower hemoglobin (*p* = 0.005), albumin (*p* = 0.011), and serum glutamic-oxaloacetic transaminase (SGOT) (*p* = 0.009) levels and more frequent use of biologic therapy (*p* = 0.009). In multivariate analysis, vitamin D deficiency remained independently associated with biologic therapy (aOR = 0.374; 95% CI: 0.148–0.946), platelet count (aOR = 0.996, 95% CI: 0.992–0.999), and SGOT (aOR = 1.05, 95% CI: 1.00–1.10), indicating consistent links between vitamin D deficiency and treatment intensity, systemic inflammation, and nutritional or metabolic status. **Conclusions:** Vitamin D deficiency is common among IBD patients and independently associates with systemic inflammation, metabolic impairment, and intensified treatment requirement, supporting its potential role as a marker of disease burden.

## 1. Introduction

Inflammatory Bowel Disease (IBD), including Crohn’s disease (CD), ulcerative colitis (UC), and indeterminate colitis (IC), represent chronic conditions of the gastrointestinal tract, characterized by periods of relapse and remission. Its pathogenesis is complex and multifactorial, involving environmental exposures, genetic susceptibility, gut microbiome alterations, and dysregulated immune responses [1,2,3]. Despite advances in treatment over recent years, including the introduction of biologic agents, disease activity remains high, underscoring the need to identify modifiable risk factors that may influence disease course [4]. Traditionally, vitamin D has been studied primarily for calcium homeostasis and bone metabolism. However, accumulating evidence indicates that the role of vitamin D extends beyond skeletal function to immune modulation, maintenance of epithelial barrier integrity, and modulation of the gut microbiota [1,5,6].

In the IBD population, vitamin D deficiency, defined as serum 25(OH)D < 20 ng/mL, is highly prevalent, with reported prevalence up to 60–80% depending on geographic region and dietary factors [5,7]. The clinical implications of this deficiency remain incompletely defined. Although several studies have associated low vitamin D levels with increased disease activity, higher relapse rates, and poor quality of life, other investigations have failed to confirm these relationships [8,9,10]. Such discrepancies may reflect differences in study design, regional variation, and patient heterogeneity.

The present study aimed to estimate the prevalence of vitamin D deficiency in a well-characterized cohort of IBD patients in Western Greece, and to explore its associations with clinical, laboratory, and therapeutic characteristics. Particular emphasis was placed on inflammatory biomarkers and the use of biologic therapy, given their clinical importance and potential links to vitamin D signaling.

## 2. Materials and Methods

### 2.1. Study Design and Participants

This cross-sectional observational study was conducted at the outpatient clinic of the Gastroenterology Division, Department of Internal Medicine, University of Patras, Greece, a major referral center for patients with gastrointestinal disorders. The patient recruitment was performed between January 2023 and December 2024. All participants were treated according to the European Crohn’s and Colitis Organization’s (ECCO) guidelines. The study included adult patients (≥18 years), with a confirmed diagnosis of IBD, which was established based on standard clinical, endoscopic, histological, and radiological criteria. Written informed consent was obtained for the collection of clinical and laboratory data.

Vitamin D levels, specifically serum 25-hydroxyvitamin D [25(OH)D], the standard clinical marker of vitamin D status, were measured in IBD patients according to the physician’s decision, based on the patient’s clinical status. To ensure representativeness, consecutive sampling was applied among IBD patients followed at the outpatient clinic. All patients with available vitamin D measurements during the recruitment period were included.

### 2.2. Data Collection

Detailed demographic and clinical information were obtained through patient interviews, review of medical records, and laboratory testing. Demographic variables encompassed age, sex, body mass index (BMI), smoking status (categorized as current, former, or never smoker), and family history of IBD, as these factors are implicated in both disease susceptibility and clinical outcomes.

Clinical variables included IBD type (CD, UC, or IC), disease duration measured in years since initial diagnosis, and the presence of extraintestinal manifestations (e.g., arthropathy, dermatological conditions, hepatobiliary complications). A history of prior IBD-related surgical procedures and hospitalizations were recorded, as both may influence long-term nutritional status, inflammatory activity, and vitamin D metabolism, and serve as markers of disease severity. Disease activity scores were calculated simultaneously with vitamin D measurement; Mayo score for UC, and both Crohn’s Disease Activity Index (CDAI) score and the Harvey–Bradshaw Index (HBI) score for CD.

Pharmacological management was also documented in detail, with particular attention to the use of amino salicylates, immunomodulators, corticosteroids, and biologic therapies, given their potential effects on systemic inflammation, immune regulation, and nutrient absorption.

Laboratory assessments were performed concurrently with vitamin D measurements using standardized hospital protocols. Hematological parameters included hemoglobin (Hb), white blood cell count (WBC), and platelet count (PLT), reflecting systemic inflammation and overall hematologic status. Inflammatory markers, including *C*-reactive protein (CRP) and erythrocyte sedimentation rate (ESR), were measured as indices of disease activity. Biochemical markers included liver enzymes [serum glutamic-oxaloacetic transaminase (SGOT), serum glutamic-pyruvic transaminase (SGPT), and γ-glutamyl transferase (γ-GT)], as well as serum albumin as a surrogate marker of nutritional and inflammatory status. Fecal calprotectin, a well-established noninvasive biomarker of intestinal inflammation, was also measured.

### 2.3. Measurement of Vitamin D Levels

Vitamin D status was assessed by measuring serum concentrations of 25(OH)D, the major circulating form and most reliable indicator of vitamin D levels, using a standardized chemiluminescent microparticle immunoassay (CMIA) [11]. This technique is used because CMIAs are validated for clinical application worldwide, offering high sensitivity, and providing reliable quantification of 25(OH)D through an automated, rapid, and highly efficient process. Furthermore, standardized CMIA platforms provide high accuracy and consistency in 25(OH)D measurements across multiple studies [11]. Vitamin D deficiency was defined as serum 25(OH)D levels < 20 ng/mL, while concentrations ≥ 20 ng/mL were considered sufficient. This threshold was based on established clinical practice guidelines and supported by epidemiological evidence [12,13,14,15].

### 2.4. Ethics

The study protocol received approval from the Ethics Committee and the Scientific Review Board of Patras University. This study conforms to the ethical principles outlined in the Declaration of Helsinki for medical research involving human subjects.

### 2.5. Statistical Analysis

Categorical variables were presented as counts (N) and corresponding percentages (%), and continuous variables were expressed as medians with interquartile ranges (IQR). Clinical characteristics and laboratory values were recorded for all patients at the time of vitamin D measurement, and disease severity scores were calculated. The Kolmogorov–Smirnov test was utilized to assess the normality of the continuous variables. The Clopper-Pearson Exact method was applied to calculate confidence intervals for the prevalence of vitamin D deficiency. Pearson’s chi-square test was used to identify potential differences in categorical variables between the examined groups. For continuous variables, group comparisons were conducted using the nonparametric Mann–Whitney U test, except for age and hemoglobin (Hb), which followed a normal distribution and were analyzed using the independent-samples *t*-test. To account for potential confounders, binary logistic regression was performed beginning with univariate subgroup analysis followed by multivariate subgroup analysis. *p*-value < 0.05 was considered to indicate statistical significance in all analyses. Data analysis was performed using IBM SPSS Statistics, version 29.0.2.0 (20) (IBM Corp., Armonk, NY, USA).

### 2.6. AI/Editing Statement

Language editing and clarity enhancement were performed using ChatGPT (OpenAI, GPT-5.1; accessed in 2025).

## 3. Results

The study comprised 184 patients with IBD. Of these, 93 (50.5%) were diagnosed with CD, 84 (45.7%) with UC, and 7 (3.8%) were classified as IC. Overall, 97 participants (52.7%) were male. The median age of the participants was 45 years (IQR: 34–61). Vitamin D deficiency, determined as serum 25(OH)D < 20 ng/mL, was identified in 67 patients, (36.4%). Detailed demographic characteristics of participants are presented in Table 1.

### 3.1. Associations Between Vitamin D Deficiency and Clinical Characteristics in IBD

Statistically significant differences in clinical and laboratory features were observed between patients with vitamin D deficiency (group 1) and those without deficiency (group 2) (Appendix A). Median hemoglobin (12.9 vs. 13.8 g/dL, *p* = 0.005) (Figure 1A), serum albumin (4.2 vs. 4.3 g/dL, *p* = 0.011) (Figure 1B), and SGOT levels (18 vs. 20 U/L, *p* = 0.009) (Figure 1C) were significantly lower in group 1 compared with group 2.

Inflammatory parameters also varied between groups; ESR value (20 vs. 12 mm/h, *p* = 0.014) (Figure 1D) and platelet count (306 vs. 264 × 10^9^/L, *p* = 0.005) (Figure 1E) were significantly higher in group 1 compared with group 2. Biologic agents were more frequently used among vitamin D deficient patients than among those with sufficient levels (86.9% vs. 73.2%, *p* = 0.009) (Figure 2). No significant differences were observed between the two groups regarding IBD type, prior IBD-related hospitalization, surgery, age, CRP, fecal calprotectin or disease activity scores.

### 3.2. Comparison of Biologic Therapy Use Between IBD Patients with and Without Vitamin D Deficiency

In multivariate analysis (Table 2), only platelet count, SGOT levels, and the use of biologic agents remained significantly associated with vitamin D status. Specifically, the use of biologic agents was more frequent among patients with vitamin D deficiency (aOR = 0.374; 95% CI: 0.148–0.946; *p* = 0.038). PLT levels were also higher in the deficiency group (aOR = 0.996; 95% CI: 0.992–0.999; *p* = 0.024). Conversely, SGOT levels were higher in patients without deficiency (aOR = 1.050; 95% CI: 1.003–1.098; *p* = 0.036).

## 4. Discussion

This single-center cross-sectional study demonstrated a remarkably high prevalence of vitamin D deficiency in IBD patients and uncovered significant associations with both inflammatory and therapeutic parameters. More than one-third of our cohort exhibited suboptimal vitamin D levels which were independently correlated to elevated platelet counts, increased ESR, and greater use of biologic therapy. These findings strongly support the concept that vitamin D deficiency may reflect a heightened inflammatory burden and a more treatment-resistant disease phenotype in IBD.

Vitamin D is increasingly recognized as a key immunomodulator in the pathogenesis of IBD. Binding to the vitamin D receptor (VDR) modulates immune responses by downregulating pro-inflammatory T helper (Th) cells, mainly suppressing Th1/Th17 pathways, and thereby reducing the secretion of cytokines, such as interferon-γ (INF-γ), interleukin 2 (IL-2), IL-17, and IL-23, which drive mucosal inflammation in IBD [10]. Furthermore, vitamin D reinforces antimicrobial peptide production, enhances epithelial barrier integrity and promotes regulatory T-cell function, while influencing the gut microbiota [10,16,17,18,19,20]. Deficiency, therefore, has been associated with increased epithelial permeability, loss of tolerance, and increased mucosal injury, ultimately contributing to a pro-inflammatory milieu [21,22]. Previous work of our research team further supports the pivotal role of the VDR pathway, showing that VDR polymorphisms, particularly ApaI, influence IBD phenotype and progression [6]. Notably, the ApaI SNP emerged as an independent predictor of IBD-related surgery, with each additional risk allele conferring an increase in adverse outcomes [6].

In the current study, vitamin D-deficient patients presented with higher platelet counts and ESR, suggesting that vitamin D deficiency is linked to inflammation activity in IBD, consistent with prior reports, indicating a correlation between low vitamin D concentrations, thrombocytosis, and pro-inflammatory state in IBD [23,24,25]. Platelet activation plays a central role in intestinal and systemic inflammation through the release of prothrombotic and immunomodulatory mediators such as platelet factor 4, transforming growth factor (TGF-β), and IL-1β, which enhance leukocyte recruitment and mucosal injury [26]. Vitamin D, in turn, exerts anti-inflammatory effects by modulating megakaryocyte proliferation and inhibiting platelet activation pathways [27,28,29]. The independent association between PLTs and vitamin D levels observed in our regression model (aOR = 0.996; 95% CI: 0.992–0.999; *p* = 0.024), supports a relationship between vitamin D and inflammation status. Similar associations have been demonstrated in large cohorts, where thrombocytosis and high platelet-to-lymphocyte ratio were predictive of vitamin D deficiency and treatment escalation in Crohn’s disease [30]. However, without longitudinal monitoring or adjustment for environmental and nutritional influences, these results should be interpreted as correlations rather than evidence of causality. The observed relationship between vitamin D levels and platelet count in our cohort may reflect a complex bidirectional interplay; however, this remains a hypothesis. Chronic inflammation could potentially decrease vitamin D concentrations through cytokine-mediated metabolic alterations, while vitamin D deficiency may in turn contribute to platelet activation and mucosal inflammation.

Whereas previous studies have frequently reported elevated liver enzymes in IBD patients, largely attributed to hepatobiliary comorbidities or drug-related hepatotoxicity [31,32], our study identified an independent association between vitamin D deficiency and lower SGOT levels. This finding is not in line with earlier observations and may reflect differences in disease phenotype and study population. Specifically, our cohort consisted of clinically stable outpatients without advanced hepatobiliary disease, in whom lower liver enzymes levels could indicate impaired nutritional or metabolic status rather than hepatocellular injury [33,34,35,36,37,38]. Given the pivotal role of the liver in vitamin D synthesis, storage, and activation, further studies incorporating hepatic imaging and metabolic profiling are needed to elucidate whether low SGOT levels reflect a response to chronic inflammation, altered vitamin D metabolism, or hepatic dysfunction in IBD.

Another important observation was the correlation between vitamin D deficiency and biologic therapy use. Since biologics are typically reserved for patients with moderate-to-severe disease, this association suggests that low vitamin D levels may not merely reflect disease activity but could also be implicated in pathways driving treatment escalation, consistent with previous reports [39]. Patients receiving anti-TNF or other biologic agents have been shown to exhibit a higher prevalence of vitamin D deficiency, whereas vitamin D deficiency has been linked to poor response to induction therapy [8,39,40]. Our logistic regression analysis confirmed this relationship, showing that patients on biologics had higher odds of vitamin D deficiency compared with those not receiving such therapy. We did not stratify the data by specific biologic agents because, during the enrollment period, many patients changed the type of biologic therapy they were receiving. Whether this highlights reverse causality or a direct biological effect remains uncertain. However, the consistent association supports the hypothesis that vitamin D has a potential supportive role in reflecting overall disease burden in IBD. Future prospective studies assessing vitamin D restoring and its impact on biologic responsiveness are warranted to determine whether correction of deficiency could improve treatment outcomes and reduce the need for therapy escalation.

Interestingly, although inflammatory markers such as CRP and fecal calprotectin were elevated in deficient patients, they did not reach statistical significance. This may be explained by the cross-sectional nature of our study and by fluctuations in inflammatory activity over time, which may affect single-point associations. Nevertheless, longitudinal studies have consistently highlighted inverse correlations between vitamin D status and systemic or mucosal inflammation, suggesting that the relationship becomes more evident when vitamin D levels and inflammatory markers are monitored dynamically [41,42].

This study has certain limitations. It was conducted at a single tertiary center, potentially limiting generalizability to broader IBD populations. Its cross-sectional design precludes causal inference, and we did not fully account for seasonal variation, patients’ diet, either as dietary intake of vitamin D or specific dietary for IBD, or vitamin D supplementation. Consequently, these limitations prevent assessment of whether correction of deficiency influences clinical outcomes. However, the findings provide an updated estimate of vitamin D deficiency prevalence in IBD patients in Western Greece, and strengthen evidence linking vitamin D status with systemic inflammation, liver function, and therapeutic intensity.

## 5. Conclusions

In conclusion, this study highlights that vitamin D deficiency is highly prevalent among patients with IBD, and is independently associated with disease activity characteristics such as systemic inflammation, altered liver enzyme patterns, and a greater need for biologic therapy. These associations suggest that low vitamin D levels may not merely reflect disease-related malnutrition or reduced sunlight exposure, but may instead represent a pattern of increased inflammatory burden and more intensive treatment requirements.

The correlations observed between vitamin D status, platelet count, and biologic use indicate that vitamin D levels may be associated with the overall inflammatory and therapeutic profile of IBD, although not necessarily as an independent marker of disease activity. Although the immunomodulatory role of vitamin D has been described in previous studies, our study did not investigate mechanistic pathways and therefore, any causal interpretations should be made cautiously. In addition, the absence of data on potential confounding factors, such as sunlight exposure, dietary habits, and supplementation, limits the ability to determine whether the observed relationships reflect causation or coexisting influences. Overall, our findings suggest that vitamin D deficiency may parallel, rather than directly drive, the inflammatory burden in IBD. Given that deficiency is common and easily correctable in IBD population, routine screening for vitamin D deficiency may hold clinical relevance. Nevertheless, it remains uncertain whether correcting vitamin D levels can modify disease activity, improve treatment outcomes, or influence the need for intensive immunosuppression. Future prospective and interventional studies are warranted to determine whether optimizing vitamin D status can influence disease activity, taking into account environmental and dietary confounding factors. Clarifying these causal relationships could ultimately support the integration of vitamin D assessment into precision-medicine oriented pathways for patients with IBD.

## Figures and Tables

**Figure 1 nutrients-17-03698-f001:**
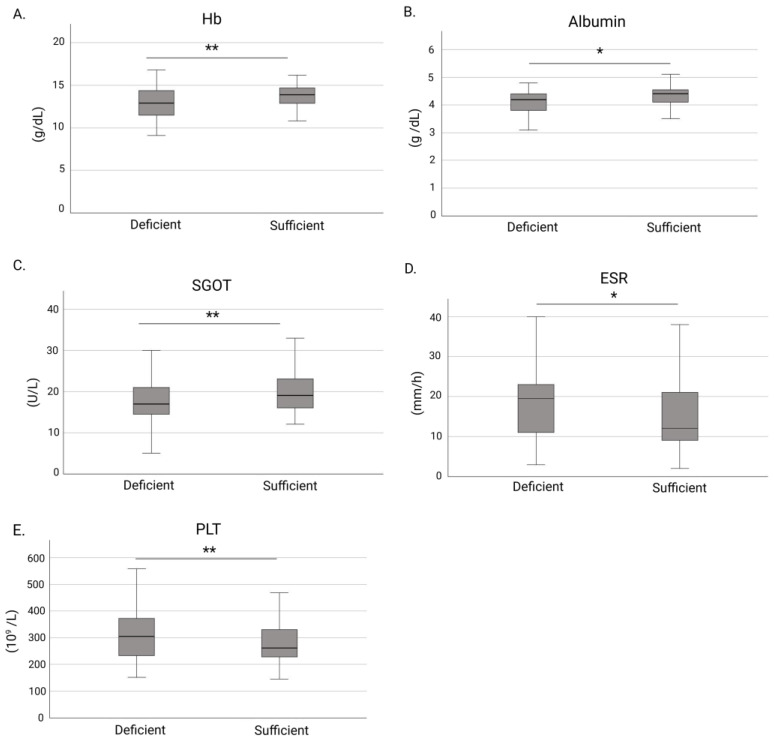
Comparisons of laboratory markers between IBD patients with and without vitamin D deficiency. (**A**) Hemoglobin levels; (**B**) Albumin levels; (**C**) SGOT levels; (**D**) ESR levels; and (**E**) Platelet count. Boxplots depict medians and interquartile ranges; whiskers indicate minimum and maximum values. Abbreviations: Hb, hemoglobin; SGOT, aspartate aminotransferase; ESR, erythrocyte sedimentation rate; PLT, platelets. * *p* < 0.05; ** *p* < 0.01.

**Figure 2 nutrients-17-03698-f002:**
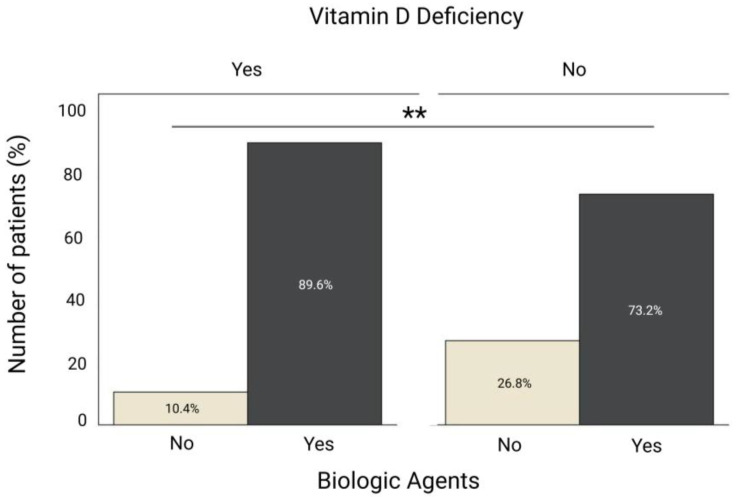
Association between vitamin D deficiency and biologic therapy use in patients with IBD. The proportion of patients receiving biologic therapy was significantly higher among those with vitamin D deficiency compared with those without deficiency. ** *p* < 0.01.

**Table 1 nutrients-17-03698-t001:** Baseline demographic characteristics of patients with IBD.

Baseline Characteristics	*N* (%)
Sex	
Male	97 (53)
Female	86 (47)
IBD type	
Crohn Disease Ulcerative Colitis	93 (50.5)
	84 (45.7)
Indeterminate Colitis	7 (3.8)
IBD family history	9 (5)
Smoking	
Current	62 (34.3)
Never	97 (53.6)
Prior	22 (12)
Vitamin D deficiency	67 (36.4)
IBD hospitalization	75 (46.9)
IBD surgery	29 (15.8)
IBD treatment	
None Amino salicylates Immunomodulators Corticosteroids Biologic Agents	7
	67
	16
	16
	142
**Baseline characteristics**	**Median (IQR)**
Age (years)	45 (34–61)
BMI	24.28 (21.70–27.77)
Vitamin D (ng/mL)	23.82 (15.95–30.58)
Mayo score (for UC)	1 (0–5.5)
CDAI score (for CD)	61.50 (29.00–103.25)
HBI score (for CD)	2 (1–4)
Hb (g/dL)	13.6 (12.2–14.6)
WBC (10^9^/L)	6825 (5700–8970)
PLT (10^9^/L)	277 (231–351)
CRP (mg/L)	0.4 (0.2–1.18)
ESR (mm/h)	15 (10–26)
SGOT (U/L)	19 (15–24)
SGPT (U/L)	19 (13–26)
γ-GT (U/L)	16 (12–26)
Albumin (g/dL)	4.2 (4.0–4.5)
Fecal calprotectin (μg/g)	194 (39–350)

Note: Percentages are calculated based on available data. The total number of cases per variable may vary due to missing values. Abbreviations: N, number; IQR, interquartile range; IBD, inflammatory bowel disease; BMI, body mass index; CDAI, Clinical Disease Activity Index; HBI, Harvey-Bradshaw Index; Hb, hemoglobin; WBC, White Blood Cell; PLT, platelets; CRP, C-reactive protein; ESR, erythrocyte sedimentation rate; SGOT, aspartate aminotransferase; SGPT, alanine aminotransferase; γ-GT, gamma-glutamyl transferase.

**Table 2 nutrients-17-03698-t002:** Univariate and multivariate logistic regression of factors associated with vitamin D deficiency in IBD patients.

Variable	Univariate Analysis	OR (95% CI)	Multivariate Analysis	aOR (95% CI)
**Age** (years)	0.515	1.006 (0.988–1.024)		
**BMI**	0.213	1.047 (0.974–1.126)		
**IBD type** **(CD vs. UC)**	0.117	1.625 (0.886–2.980)		
**IBD surgery**	0.188	0.580 (0.257–1.306)		
**Biologic Agents**	**0.012**	0.319 (0.131–0.775)	**0.038**	0.374 (0.148–0.946)
**Number of Biologic Agents**	0.217	0.791 (0.545–1.148)		
**Hb** (g/dL)	**0.008**	1.289 (1.070–1.553)	0.317	1.097 (0.915–1.314)
**WBC** (10^9^/L)	0.611	1.000 (1.000–1.000)		
**PLT** (10^9^/L)	**0.001**	0.994 (0.991–0.998)	**0.024**	0.996 (0.992–0.999)
**CRP** (mg/L)	0.068	0.873 (0.754–1.010)		
**ESR** (mm/h)	0.057	0.982 (0.964–1.001)		
**SGOT** (U/L)	**0.010**	1.061 (1.014–1.109)	**0.036**	1.050 (1.003–1.098)
**SGPT** (U/L)	0.284	1.012 (0.990–1.035)		
**Albumin** (g/dL)	0.473	1.053 (0.919–1.170)		
**Mayo score**	0.094	0.889 (0.775–1.020)		
**CDAI score**	0.748	0.999 (0.992–1.006)		
**HBI score**	0.185	0.916 (0.805–1.043)		

Note: Odds ratios (ORs) are expressed with group 2 (IBD patients without vitamin D deficiency) as the reference category. Abbreviations: OR, Odds Ratio; CI, Confidence Interval; aOR, adjusted Odds Ratio; BMI, body mass index; IBD, inflammatory bowel disease; CD, Crohn’s disease; Hb, hemoglobin; WBC, White Blood Cell; PLT, platelets; CRP, *C*-reactive protein; ESR, erythrocyte sedimentation rate; SGOT, aspartate aminotransferase; SGPT, alanine aminotransferase; CDAI, Clinical Disease Activity Index; HBI, Harvey-Bradshaw Index.

## Data Availability

The data presented in this study are available on request from the corresponding author. The data are not publicly available as they involve human subjects, and their confidentiality and ethical considerations must be respected.

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
