# Peer review of "High Prevalence and Clinical Associations of Vitamin D Deficiency in Inflammatory Bowel Disease: Evidence from a Tertiary Center Cohort"

_nutrients, 2025, doi:10.3390/nu17233698_

Round 1

Reviewer 1 Report

Comments and Suggestions for Authors

It is certainly useful to know any nutritional deficiencies as well as other clinically relevant abnormalities that may be prevalent among our patients. In this study, the authors have called attention to the common occurrence of vitamin D deficiency and demonstrated a certain correlation with other inflammatory biomarkers. This observation is useful in clinical terms, especially with regard to the possible benefits of vitamin D supplementation in such circumstances.

Speculation, however, regarding the incremental benefits of vitamin D measurement per se as an independent marker of disease activity [lines 378-80] are not supported by the data; nor, in the absence of analyzing confounding factors (e.g., sun exposure, dietary habits, supplemental vitamin intake [acknowledged in lines 364-66]), is extensive theorizing justified about the role of vitamin D in the underlying pathophysiology of IBD. The lengthy discussion of these theories and numerous corresponding references could well be curtailed.

A couple of editorial points: (1) Five of the lines in Table 1 do not require yes/no breakdown;     (2) Table 2 might better be relegated to supplementary data.

Author Response

Reviewer 1

It is certainly useful to know any nutritional deficiencies as well as other clinically relevant abnormalities that may be prevalent among our patients. In this study, the authors have called attention to the common occurrence of vitamin D deficiency and demonstrated a certain correlation with other inflammatory biomarkers. This observation is useful in clinical terms, especially with regard to the possible benefits of vitamin D supplementation in such circumstances.

Response to Reviewer 1: We thank reviewer 1 for his positive feedback on our manuscript. 

Comment 1: Speculation, however, regarding the incremental benefits of vitamin D measurement per se as an independent marker of disease activity [lines 378-80] are not supported by the data; nor, in the absence of analyzing confounding factors (e.g., sun exposure, dietary habits, supplemental vitamin intake [acknowledged in lines 364-66]), is extensive theorizing justified about the role of vitamin D in the underlying pathophysiology of IBD. The lengthy discussion of these theories and numerous corresponding references could well be curtailed.

Response to comment 1: We thank the Reviewer for this important comment. In the revised manuscript, we have substantially reduced the mechanistic information and removed redundant references from the Discussion section to avoid overinterpretation of pathways not directly assessed in our study (page 8, lines 286-287, page 9, lines 296-297 & 300-309 & 317-325 & 341-343). We have also rewritten the Conclusion section to eliminate any implication that vitamin D may act as an independent marker of disease activity and to ensure that our interpretations remain appropriately cautious and aligned with the presented data.

Comment 2: A couple of editorial points:

(1) Five of the lines in Table 1 do not require yes/no breakdown; 

Response to comment 2: Table 1 has been modified in the revised manuscript.

Comment 3: (2) Table 2 might better be relegated to supplementary data.

Response to comment 3: Table 2 has been moved to the supplementary materials and is now titled Supplementary Table 1.

Comment 4: The conclusions must be improved.

Response to comment 4: We thank the Reviewer for this comment. The Conclusion section has been revised to improve clarity, strengthen the key messages, and ensure that the interpretations are fully aligned with our data.

Comment 5: The figures and tables can be improved.

Response to comment 5: We thank the Reviewer for this comment. In the revised manuscript both figures and tables have been improved.

Reviewer 2 Report

Comments and Suggestions for Authors

General Comments

The authors have presented a very well-written paper discussing the relationship between vitamin D deficiency and inflammatory bowel disease. The authors have clearly described the research gap and explained where their research fits into the field. The methods are sufficiently described. The authors have also sufficiently described the results and stated the clinical relevance of their study, and its potential clinical translation. There are only some minor comments regarding grammar mistakes, and the figures could benefit from minor changes and improvements.

Minor Comments

Abstract

  • Line 15: do the authors mean Vitamin D or active vitamin D (i.e. 1,25(OH)2D3 plays a role in immune regulation?
  • Line 17: Missing comma, see revision below
    • The aim of the study is to assess the prevalence of vitamin D deficiency in a well-characterized IBD cohort in Western Greece, and explore its associations with clinical features, laboratory biomarkers, and treatment intensity.
  • Line 24: the authors could briefly define 25(OH)D in line 24 – active form, inactive form etc.

Introduction

  • Line 56: please re-define VD deficiency threshold (< 20 ng/mL).

Materials and methods

  • Line 117: please reference the standardised immunoassay.
  • Line 80: the author should define what markers they were specifically measuring, vitamin D is too vague and needs further clarification.
  • Line 91: the square brackets “[CD, UC, or IC]” may not be necessary here, normal brackets will suffice.
  • Did the authors consider the patients’ diets as a clinical parameter? Especially since IBD patients will often be on specific diets.

Results

  • Line 144: the authors should use numerals for 184, and could potentially rephrase the sentence to read “the study comprised of 184 patients with IBD”.
  • Table 1: if the units are mentioned in the column title, the authors do not need to repeat the unit in the data.
  • Line 170: there is an error in the text, the authors state“…were significantly lower in group 1 compared with group”. Please correct this and state which group this is being compared to.
  • Figure 1:
    • The axes and especially labels are much too small – please increase the font size.
    • The X-axis title is not required (“vitamin D deficiency”) and can be removed. The author should also remove “vit D” abbreviation, not required
      • If the authors want to keep this axis title, then the groups could read “deficient” and “sufficient” with x-axis labelled as “vitamin D status”.
    • There are missing units on the Y-axes – the marker is not sufficient. As a suggestion, the marker could be the graph title, and the y-axis could read concentration with the unit of measurement.
  • Figure 2:
    • The author should rename the y-axis title to “number of patients (%)”
    • The authors should consider stratifying the data for the different biologic agents, instead of generalising “biologic agent” vs. “no biologic agent”, or explain in the results/discussion why this was not possible.
    • Please remove the units in the y-axis and put the unit in the y-axis title, as suggested above.
  • Table 2: as previously mentioned, the author does not need to put % in the data in the table when the units are already stated in the titles of each column. The author could be more throughout table 2, e.g. “Gender” column = not used % in the table, but for the rest, have used the % symbol
  • Title 3.2: the author has consistently used title case for subheadings and headings, but not on title 3.2. Please keep this consistent.

Discussion

  • Lines 288-291: the author could consider moving this information to the introduction.
  • Lots of new information from lines 287-300: maybe try to incorporate some into introduction as lots of new information not previously mentioned.
  • Line 301: small typing error: missing – between vitamin D and deficient à vitamin D-deficient
  • Line 363: missing comma, see revised sentence below
    • It was conducted at a single tertiary center, potentially limiting generalizability to broader IBD populations
  • Line 373: missing comma, see revised sentence below
    • These associations suggest that low vitamin D levels may not merely reflect disease-related malnutrition or reduced sunlight exposure, but rather represent a biochemical marker of higher inflammatory activity and treatment intensity.

Author Response

Reviewer 2

The authors have presented a very well-written paper discussing the relationship between vitamin D deficiency and inflammatory bowel disease. The authors have clearly described the research gap and explained where their research fits into the field. The methods are sufficiently described. The authors have also sufficiently described the results and stated the clinical relevance of their study, and its potential clinical translation. There are only some minor comments regarding grammar mistakes, and the figures could benefit from minor changes and improvements.

Response to Reviewer 1: Thank you very much for your positive feedback on our manuscript. 

Comment 1: Line 15: do the authors mean Vitamin D or active vitamin D (i.e. 1,25(OH)2D3 plays a role in immune regulation?

Response to comment 1: In the revised manuscript, we now clarify that the immunoregulatory effects refer to the active form of vitamin D, 1,25-dihydroxyvitamin D₃ [1,25(OH)₂D₃] (page 1, lines 16-17).

Comment 2: Line 17: Missing comma.

Response to comment 2: We thank the reviewer for this comment. We have now revised this section as suggested (page 1, line 20).

Comment 3:  Line 24: the authors could briefly define 25(OH)D in line 24 – active form, inactive form etc.

Response to comment 3: We have revised as suggested (page 1, lines 24-26).

Comment 4: Line 56: please re-define VD deficiency threshold (< 20 ng/mL).

Response to comment 4: We thank the reviewer who pointed out this important omission. We have revised this section to include the threshold for VD deficiency < 20ng/ml (page 2, line 59).

Comment 5: Line 117: please reference the standardized immunoassay.

Response to comment 5: We thank the reviewer for this comment, and we have clarified in the revised manuscript the method of standardized immunoassay (page 3, lines 121-126)

Comment 6: Line 80: the author should define what markers they were specifically measuring, vitamin D is too vague and needs further clarification.

Response to comment 6: We have added the specific marker that was measured, as suggested (page 2, lines 83-84).

Comment 7: Line 91: the square brackets “[CD, UC, or IC]” may not be necessary here, normal brackets will suffice.

Response to comment 7: We thank the reviewer for this comment. We have now revised this section as suggested (page 3, line 95).

Comment 8: Did the authors consider the patients’ diets as a clinical parameter? Especially since IBD patients will often be on specific diets.

Response to comment 8: We thank the reviewer for this comment. We have added clarification to the discussion section where we cited the limitations of our study. Specifically, we mention that the patients' dietary habits were not considered (page 10, lines 357).

Comment 9: Line 144: the authors should use numerals for 184 and could potentially rephrase the sentence to read “the study comprised of 184 patients with IBD”.

Response to comment 9: We have now revised the sentence as requested (page 4, line 157).

Comment 10: Table 1: if the units are mentioned in the column title, the authors do not need to repeat the unit in the data.

Response to comment 10: We thank the reviewer for this suggestion. In the revised manuscript, we have modified Table 1, as suggested.

Comment 11: Line 170: there is an error in the text, the authors state“…were significantly lower in group 1 compared with group”. Please correct this and state which group this is being compared to.

Response to comment 11: We thank the reviewer for bringing this to our attention. We have added group 2, in the revised manuscript (page 5, line 186).

Comment 12: Figure 1: The axes and especially labels are much too small – please increase the font size.

Response to comment 12: Thanks for the notice, the font size has been increased.

Comment 13: The X-axis title is not required (“vitamin D deficiency”) and can be removed. The author should also remove “vit D” abbreviation, not required

Response to comment 13: Thanks for the notice, we have revised as suggested.

Comment 14: If the authors want to keep this axis title, then the groups could read “deficient” and “sufficient” with x-axis labelled as “vitamin D status”.

Response to comment 14: Thanks for the notice, we have revised as suggested.

Comment 15: There are missing units on the Y-axes – the marker is not sufficient. As a suggestion, the marker could be the graph title, and the y-axis could read concentration with the unit of measurement.

Response to comment 15: Thanks for the notice, we have revised as suggested.

Comment 16: Figure 2: The author should rename the y-axis title to “number of patients (%)”

Response to comment 16: Thanks for the notice, we have revised as suggested.

Comment 17: The authors should consider stratifying the data for the different biologic agents, instead of generalising “biologic agent” vs. “no biologic agent”, or explain in the results/discussion why this was not possible.

Response to comment 17: Thanks for the interesting comment. We had considered this stratification; however, during the enrollment period, many patients receiving biologic agents changed treatment type, making such an analysis difficult to perform. This limitation is now explained in the Discussion (page 9, lines 338-339).

Comment 18: Please remove the units in the y-axis and put the unit in the y-axis title, as suggested above.

Response to comment 18: Thanks for the notice, we have revised as suggested.

Comment 19: Table 2: as previously mentioned, the author does not need to put % in the data in the table when the units are already stated in the titles of each column. The author could be more throughout table 2, e.g. “Gender” column = not used % in the table, but for the rest, have used the % symbol

Response to comment 19: We thank the Reviewer for this comment. We have now revised Table 2 to ensure consistency and removed the percentage symbols (%) within the table cells.

Comment 20: Title 3.2: the author has consistently used title case for subheadings and headings, but not on title 3.2. Please keep this consistent.

Response to comment 20: We thank the reviewer for bringing this to our attention. We have revised as suggested (page 7, lines 242-243).

Comment 21: Line 288-291: the author could consider moving this information to the introduction.

Response to comment 21: We thank the Reviewer for this suggestion. We respectfully believe that the mechanistic description of vitamin D’s immunoregulatory role, which includes VDR-mediated pathways, T-cell modulation, cytokine regulation, and barrier integrity, is too detailed for the Introduction. The mechanistic content presented in lines 288–291 in the Discussion provides biological context for the associations observed in our study and supports the interpretation of our findings. For this reason, we have retained this information in the Discussion section.

Comment 22: Lots of new information from lines 287-300: maybe try to incorporate some into introduction as lots of new information not previously mentioned.

Response to comment 22: We thank the Reviewer for this helpful suggestion. While we appreciate the recommendation to incorporate part of the material from lines 287–300 into the Introduction, we believe that the content in this section is primarily mechanistic and detailed (e.g., VDR-related pathways, cytokine regulation, T-cell modulation). Such information may be too specific for the Introduction, which aims to provide a concise clinical rationale for the study. The added mechanistic details are more appropriately placed in the Discussion, where they support the interpretation of our findings and offer biological context for the observed associations. Therefore, we have retained this material in the Discussion section.

Comment 23: Line 301: small typing error: missing – between vitamin D and deficient à vitamin D-deficient

Response to comment 23: We have revised as suggested.

Comment 24: Line 363: missing comma.

Response to comment 24: We have revised as suggested.

Comment 25: Line 373: missing comma.

Response to comment 25: We have revised as suggested.

Round 2

Reviewer 1 Report

Comments and Suggestions for Authors

The authors have softened some of their speculations regarding the role of vitamin D deficiency per se influencing the course of the disease, Hence, their paper now stands on stronger scientific ground.